# Imaging Features of Intraosseous Schwannoma: A Case Series and Review of the Literature

**DOI:** 10.3390/diagnostics13091610

**Published:** 2023-05-02

**Authors:** Firoozeh Shomal Zadeh, Arash Azhideh, Jose G. Mantilla, Vijaya Kosaraju, Nitin Venugopal, Cree M. Gaskin, Atefe Pooyan, Ehsan Alipour, Majid Chalian

**Affiliations:** 1Division of Musculoskeletal Imaging and Intervention, Department of Radiology, University of Washington, Seattle, WA 98915, USA; 2Department of Laboratory Medicine and Pathology, University of Washington, Seattle, WA 98915, USA; 3Division of Musculoskeletal Imaging and Intervention, Department of Radiology, University Hospitals Cleveland Medical Center, Case Western Reserve University, Cleveland, OH 44106, USA; 4Division of Musculoskeletal Imaging and Intervention, Department of Radiology, University of Virginia, Charlottesville, VA 22903, USA

**Keywords:** schwannoma, intraosseous, imaging, MRI, CT, radiograph

## Abstract

To characterize the imaging features of patients with pathologically confirmed intraosseous schwannoma (IOS), institutional pathology and imaging databases were searched for IOS cases over a period of 17 years. A musculoskeletal radiologist evaluated all imaging studies. Additionally, a literature search was performed to identify IOS cases that had imaging findings of at least two modalities. Six patients (one female, five males, mean age of 50 ± 14 years) with IOS were identified, with all lesions localized to the lumbosacral region. Radiographic imaging was available in four patients, while all patients underwent CT and MR imaging. Radiographs depicted lytic lesions, and CT depicted heterogeneous expansile lesions with centrally hypodense areas and peripheral sclerosis. All cases involved extra-osseous extension, producing a mass effect on adjacent soft tissues and nerve roots. On MRI, the neoplasms displayed iso- to- slightly- low signal intensity on T1-weighted images and hyperintense signal intensity on T2-weighted images with heterogeneous enhancement. The literature review resulted in 102 IOS cases, which to the best of our knowledge, is the largest review on IOS, and the imaging findings of the previously published cases were the same as our cases. IOSs are rare benign neoplasms that should be considered in the differential diagnosis of well-defined expansile lytic lesions with sclerotic borders. This is particularly important in middle-aged adults with mandibular, sacral, or vertebral body mass.

## 1. Introduction

Schwannomas are the most common neoplasms arising from Schwann cells of the nerve sheaths. These tumors are benign neoplasms, which more likely arise from peripheral sensory nerve axons since sensory nerves are surrounded by more Schwann cells than motor axons [1,2]. Schwannomas are predominantly present in the skin or subcutaneous tissue of the head and neck regions and the flexor aspects of the extremities. No significant risk factors are associated with schwannomas, although exposure to ionizing radiation was implicated in one previous report, and schwannomas are reported to be associated with NF-2 syndrome [3].

Intraosseous schwannomas (IOSs) are exceedingly rare, comprising less than 0.2% of primary bone tumors [1,4]. These tumors are slow-growing benign neoplasms originating within the medullary cavity of bones or from adjacent structures. The three most accepted theories explaining the intraosseous component of these IOSs are: (1) intraosseous tumor arising purely in the medullary cavity with rarefaction of the bones, (2) intraosseous tumor originating in the nutrient canal as dumbbell-shaped mass causing canal enlargement, or (3) tumors arising from adjacent extraosseous structures eroding into the bone [5,6]. IOSs mainly arise in mandible, where the mandibular nerve is predominantly composed of sensory nerve fibers, and sacrum, as the sacral foramina contains many sensory nerves. Other affected locations include maxilla, spine, petrous apex, and less commonly, the long bones of the extremities [5,6,7]. IOSs have been associated with NF-1 and Carney syndromes [7,8]. The diagnosis was not initially suspected in many reported IOS cases until histopathological studies of the biopsied lesions, most likely due to the rare incidence of this benign neoplasm [9].

Given the low prevalence, this diagnosis of IOS is not typically included in the differential considerations of lytic bone lesions, which practically include more common skeletal disease processes such as osteomyelitis or bone metastases [5,10]. A definitive diagnosis of IOS is currently made by histopathologic examination, which typically identifies spindle cells with alternating cellular and myxoid regions [1]. In this manuscript, we characterize the clinical and radiological features of six patients with IOS and conduct a literature review focused on radiologic features. To our knowledge, the prior literature regarding IOSs includes small case series and case reports. In this case series and literature review, we aimed to conduct the imaging appearance of IOSs to help differentiate them from other lytic lesions of the bone.

## 2. Materials and Methods

Our institutional imaging and pathology databases were retrospectively searched for patients with intraosseous schwannoma over a period of 17 years (2001–2018). Ethical review and approval were waived for this study, as it was a retrospective study with no identifiable information of patients. The diagnosis of IOS was confirmed in all cases via histopathological analysis following surgical or core needle biopsy of the lesions. Clinical, demographic, and pathologic information available in the electronic medical record was reviewed.

All relevant available imaging studies were reviewed by a fellowship-trained musculoskeletal radiologist using a PACS (picture archiving and communication system) workstation (Vue PACS, Carestream Health, Rochester, New York). A typical combination of MRI sequences included multiplanar T1-weighted spin-echo (SE), short tau inversion recovery (STIR), and pre- and post-contrast T1-weighted fat-suppressed sequences. Diffusion-weighted imaging was not part of the routine institutional tumor protocol at the time of imaging. MRI was performed by using either a 1.5-T (Avanto, Magento Symphony or Magento Vision Plus; Siemens or Intera, Philips Healthcare) or a 3.0-T (Verio; Siemens) system with phased-array coils and patients in a supine position.

We also reviewed the published English literature between 1954 and 2021, using the US National Library of Medicine database through the National Institutes of Health (PubMed), Scopus, Embase, and Web of Science. The search terms utilized were intraosseous schwannoma, intraosseous neurilemmoma, intra-osseous schwannoma, and intra-osseous neurilemmoma. There was no restriction on the date of publication. All research studies were selected in each database. All studies were cross-referenced between the three databases, and duplicate studies were removed. Studies with radiological findings for IOSs through at least two imaging modalities were included. Studies that only included radiographic imaging were excluded, as we were interested in the imaging presentation of IOSs among various imaging modalities such as CT and MRI. Additionally, any studies that were not specific to IOS, only had an abstract with no full text available, or lacked detailed imaging findings, were excluded. A flowchart summarizing the study selection and exclusion process is depicted in Figure 1. We used SPSS version 24 (Armonk, NY, USA) to conduct analyses and pool the data extracted from the literature.

## 3. Results

### 3.1. Case Series

#### 3.1.1. Demographic Characteristics

A retrospective review of institutional imaging and pathology databases resulted in the identification of six patients (five male and one female) with confirmed intra-osseous schwannoma. There was a significant male predilection (5/6, 83%). The mean age at presentation was 50 years (ranging from 28 to 64 years with a standard deviation of 14 years). Regarding presenting symptoms, 67% of patients (4/6) reported pain, including lower back and leg pain, 17% (1/6) reported difficulty walking, and 33% (2/6) reported sudden onset foot drop. Lesions ranged from 1.9 cm to 12 cm in size and were located in the lumbosacral region as follows: L5 (*n* = 1). S1 (*n =* 1), S2 (*n =* 1), S1 and S2 (*n =* 1), S1-S3 (*n =* 1), and S2 and S3 (*n =* 1). Details including the size and location of the lesions, demographic information of the patients, and the imaging features are reported in Table 1.

#### 3.1.2. Radiographic Imaging

Radiographs were available on four patients. One of these only had a postoperative radiograph in which the lesion was occult. Other radiographs demonstrated geographic lytic lesions with sclerotic borders (Figure 2A,B).

#### 3.1.3. Computed Tomography

All patients had CT images available. Four patients underwent diagnostic CT studies, while the other two had preprocedural CT images obtained during the CT-guided biopsy. Imaging findings mostly included expansile intraosseous lytic lesions extending into adjacent spaces, including the neuroforamina, epidural space, and central spinal canal (Figure 2C,D). There was one case of a locally aggressive tumor extending anteriorly into the pelvis and posteriorly into the epidural space (Figure 3). The masses were characterized on imaging as heterogeneous, expansile lesions with centrally hypodense areas and peripheral rims of sclerosis.

#### 3.1.4. Magnetic Resonance Imaging

All IOSs demonstrated consistent signal characteristics on MRI with a low-to-intermediate signal intensity to muscle on T1-weighted images (T1-WI) and heterogenous hyperintense signal intensity on fluid-sensitive sequences. Extra-osseous extension was frequently demonstrated, including invasion into neural foramina with displacement of adjacent nerve roots (Figure 2E,F) and involvement of the epidural space. There was posterior osseous extension from vertebral bodies into the pedicles and spinous processes. There was avid post-contrast enhancement with scattered areas of central non-enhancement.

#### 3.1.5. Histologic Findings

The characteristic features of schwannoma include the presence of short fascicles of neoplastic cells with areas of nuclear palisading (Antoni A), alternating with paucicellular Antoni B areas. No significant nuclear pleomorphism, conspicuous mitotic activity, or other features of malignancy were identified. The transition between tumor and the involved bone showed areas of bone resorption and reactive features (Figure 4). These tumors had diffuse nuclear and cytoplasmic reactivity for S100 (not shown).

### 3.2. Review of the Literature

#### 3.2.1. Study Selection

We conducted a literature search to identify IOS cases, which described their clinical and radiological features. Our search resulted in 188 PubMed, 447 Embase, 303 Scopus, and 155 Web of Science research articles. All abstracts were reviewed, and only IOS cases with imaging findings were selected (*n =* 255). Additional studies were filtered out based on the following exclusion criteria: Duplicate articles (*n =* 30), articles that only had an abstract available and the full text could not be accessed (*n =* 35), studies that did not have at least two imaging modalities of IOS or did not have detailed imaging findings (*n =* 74), malignant peripheral nerve sheath tumors (*n =* 8), not specific to IOS (*n =* 15), commentary articles (*n =* 3), recurrence (*n =* 1), and studies that did not have individual clinical and radiological findings for each reported case (*n =* 3). Finally, 86 articles with 102 cases were further reviewed, and a summary of their various clinical findings, tumor locations, and imaging features were extracted.

#### 3.2.2. Study Characteristics

The average age of patients (*n =* 102) at the time of presentation was 42 years, with a standard deviation of 18 years and a range from 3 to 87 years. The gender distribution was relatively balanced, with 53 (52%) female and 49 (48%) male patients. We categorized the IOSs based on their anatomic location; a total of 34.3% (35/102) of cases were found in the head and neck (Table 2), 34.3% (35/102) were located in the trunk (Table 3), and 31.4% (32/102) were in the extremities (Table 4). Among the IOSs of the extremities, 34% (11/32) were found in the upper limb and 66% (21/32) were in the lower limb. The most frequently affected anatomic locations were the mandible in (18/102), sacrum (17/102), and vertebral bodies (16/102). The distribution of cervical (6/102), thoracic (5/102), and lumbar (5/102) vertebral body involvement was relatively equal.

Regarding the initial presentation, most of the patients presented with slow growing mass resulting in swelling and or pain. Four patients with lesions in petrous apex presented with a compression mass effect on adjacent cranial nerves causing double vision, hearing loss, and decreased facial sensation. Rozman et al. reported a case of IOS in petrous apex, which presented with intermittent vertigo, tinnitus, facial numbness, hearing loss, diplopia, and ataxia [11]. One IOS in the spheno-orbital region was diagnosed after proptosis and lateral gaze impairment [17]. Schreuder et al. reported a patient with IOS C6 vertebral body that was presented with neck pain and dysphagia [43]. Six patients with IOS of vertebral bodies (T9 and L4) and sacrum demonstrated claudication, gait disturbance, and back pain.

We found 11 cases diagnosed after pathological fracture due to IOS. Nooraie et al. reported a case of T12 IOS, which was detected after a burst fracture in the T12 vertebral body following a car accident [53]. IOS was also discovered as an incidental finding in 14 cases. Mizuno et al. reported a case of a mass in the sacrum that was found incidentally during a routine physical examination and later diagnosed as IOS following biopsy [44].

#### 3.2.3. Imaging Findings

The majority of IOS cases in the literature review had radiographic findings similar to those in our patients. Most IOSs on radiographs were well-defined, expansile, and lytic (96%) lesions with sclerotic borders (72%). The lesions had narrow zones of transition. A periosteal reaction was seen in four cases, which was associated with a pathologic fracture in two of them [65,72,75,82]. Typical CT findings included heterogenous, lytic lesions with sclerotic margins suggestive of the slow-growing and benign nature of these lesions. Peripheral and/or mild scattered central calcifications have been reported in four cases [42,46,62,84]. Cortical destruction and endosteal scalloping were also seen in 61% of cases. On MRI, IOSs generally demonstrated isointense to slightly hypointense signal on T1WI (94%) with heterogeneous high signal intensities on T2-weighted images (T2WI) (98%). Post-contrast images demonstrated enhancement (93%) of the lesions, although the enhancement pattern was not uniform and differed from mild homogenous to intense peripheral enhancement. Extra-osseous soft tissue components were reported in large lesions with cortical breakthrough [48,63].

## 4. Discussion

Schwannomas, originating from Schwann cells, are benign neoplasms that more likely arise from sensory nerve axons since they are surrounded by more Schwann cells than motor nerves [2]. This tumor predominantly presents in the soft tissues of the head and neck regions, and schwannomas of the bone are extremely rare. In this case series, we describe the imaging and clinical findings of six cases derived from the review of local institutional images. We combined our findings with those from previous cases in the literature to better characterize the radiologic and clinical presentation of these tumors. Clinically, IOSs are benign neoplasms that are discovered on imaging studies incidentally or after patients present with symptoms such as pain and swelling. Symptoms, if present, often have a slow onset and likely persist for many years before diagnosis [84]. The peak incidence of this type of neoplasm is in the fourth-to-sixth decade of life, with no predilection for gender or race [10]. Their presenting symptoms are often localized pain or swelling, with a few cases of sensory and motor impairments reported [8,11]. IOSs could affect nearly all the bones but the most frequent reported locations are the mandible and sacrum. Some other reported locations include the vertebrae, cranium, scapula, sternum, ribs, femur, tibia, ulna, and digital phalanges [5,7,8,9].

In our literature review of 102 published IOS case reports, the average age of the patients was 42 years. Meanwhile, our case series of six patients had an average age of 50. Both age averages are within the reported age range for IOS, which is in the fourth-to-sixth decade of life. Additionally, although the distribution of male to female patients was relatively even in the literature review (53 female versus 48 male patients), our case series includes only 1 female patient and 5 male patients. Our reported cases were all in the lumbosacral region, while our literature review had lesions in the appendicular and axial skeletons. The discrepancy in age and gender distribution is most likely due to the small sample size in our case series. The uniformity of IOS localizing to the lumbosacral region in our series is likely due to the small sample size and some predilection for this location. Although it has been reported that the mandible is the most common location for IOSs, the majority of mandibular IOS cases were excluded from our review due to only having one imaging modality (radiographic) available.

Intraosseous schwannomas display benign imaging features on radiographs, including an osteolytic pattern with a narrow zone of transition, a thin peripheral sclerotic rim, variable osseous expansion, and minimal periosteal reaction [7]. Even though the radiological findings are not particularly helpful in narrowing the differential, they suggest the benign nature of the mass. Due to the nonspecific characterization on radiographic imaging, it is hard to differentiate these neoplasms from other bone lesions [71]. In our case series, IOSs were either occult on radiographs or visualized as lytic lesions with sclerotic borders. CT imaging of our reported IOS cases revealed lytic, lobulated, intraosseous masses extending through the cortex into the paraspinal soft tissue or adjacent neural foramina and nerve roots.

On MRI studies, IOSs have been reported to present as lesions with isointense to slightly hypointense signal to muscle on T1WI and heterogenous hyperintense signal on T2WI [5,92]. They typically have similar MR imaging features to soft tissue schwannomas. On T2WI, soft tissue schwannomas have a characteristic “target” sign, which is attributed to the central distribution of hypercellular Antoni type A components and peripheral distribution of hypocellular Antoni type B components [5,93]. Schwannomas typically show a heterogeneous pattern of enhancement, which is more pronounced in the periphery, although homogeneous enhancement is also observed [17,33,41,62]. In our literature review, IOSs in different anatomical locations were reported to have similar signal intensities with various patterns of enhancement. Similarly, our six reported IOS cases displayed a homogeneously low-to-intermediate signal intensity on T1WI and heterogeneous high signal intensity on T2WI with heterogeneous enhancement. We did not observe the more peripheral predominant enhancement pattern in our case series, of which is reported in soft tissue schwannomas. In addition, the characteristic “target” sign signal intensity pattern of soft tissue schwannoma was not observed in our case series. Few reported cases in our literature search exhibited such a characteristic pattern of signal intensity [5,18,35].

Other advanced imaging techniques such as positron emission tomography (PET) and diffusion-weighted imaging (DWI) may depict more characteristic findings and help differentiate these benign neoplasms from malignant ones. Tamura et al. reported that on diffusion-weighted imaging, IOSs demonstrate a high signal intensity on low b value images without diffusion restriction [13]. PET studies have been typically used to assess tumor metabolism in malignant neoplasms; however, significant data are not available on the efficacy of PET scans in benign soft tissue tumors [94]. Preliminary studies indicate that using [7] deoxy-fluoro-D-glucose (FDG), the most commonly used radiolabeled tracer for PET studies, can help in differentiating benign vs. malignant musculoskeletal soft tissues and bone tumors [95]. Kashima et al. reported that on a PET scan, the IOS neoplasm was mildly FDG avid [79]. Additionally, in another two cases, an increased radioisotope uptake at the site of the IOS lesion was reported [49,96]. However, Beaulieu et al. found that PET scans are not clinically helpful in identifying schwannomas since FDG uptake is variable and a clear cut-off value cannot be established [94].

IOSs are generally benign and only one case of malignant transformation in a spinal IOS 2 years after subtotal resection has been reported [40]. However, there have been a few reported cases of primary intraosseous malignant peripheral nerve sheath tumors (MPNSTs) in the literature [4,7]. These malignant tumors have been reported in mandible, maxilla, femur, cervical, and thoracic spine [40,41,42,43,44,45,46,47,48,49,50,51,52,53,54,55,56,57,58,59,60,61,62,63,64,65,66,67,68,69,70,71,72,73,74,75,76,77,78,79,80,81,82,83,84,85,86,87,88,89,90,91,92,93,94,95,96,97,98]. The most significant risk factor for MPNSTs is NF-1 syndrome, as 25–50% of MPNSTs develop in NF-1 patients and 4.6% of NF-1 patients develop MPNSTs [99,100]. These malignant tumors often appear sporadically and spontaneously [101]. One reported case described a 62-year-old woman with a history of right midfoot pain who was diagnosed with primary intraosseous MPNST of medial cuneiform after biopsy [102]. She was ultimately treated with neoadjuvant radiotherapy followed by wide local excision and allograft reconstruction. Currently, there is no evidence to suggest that MPNSTs are associated with NF-2 syndrome. It is highly unlikely for patients with this syndrome to have malignant transformation of their schwannomas to MPNSTs [101].

Due to the nonspecific presentation of this type of neoplasm, the diagnosis of IOS is not typically suspected or confirmed until after histopathological studies. Typically, schwannomas are well-circumscribed neoplasms composed of cytologically bland spindle cells arranged in short fascicles with areas of nuclear condensation (Antoni A) alternating with paucicellular areas (Antoni B). In longstanding cases, degenerative changes, such as myxoid change, hyalinization, nuclear enlargement, and hemosiderin deposition, may obscure the characteristic morphologic features. Essentially, all of these lesions have diffuse nuclear and cytoplasmic reactivity for S100 protein [10]. This rare benign neoplasm is associated with a good prognosis, and there has been only one report of malignant transformation [103]. Curettage and bone grafting are the treatment of choice for this type of schwannoma. In a series of 31 cases, there was a 16% recurrence rate following incomplete excision, but tumor recurrence was not observed in cases that had undergone complete excision [104].

This study is limited by the small sample size related to the rare nature of the condition. Although the most common location for IOSs is reported to be the mandible, that was not our finding in this case series as all of our cases were located in the lumbosacral region. Our study is also limited by the lack of advanced MR sequences such as diffusion and perfusion imaging at the time of examination and advancements in imaging protocol, as cases were collected over a period of 17 years.

## 5. Conclusions

We evaluated 6 cases of sacral intraosseous schwannoma and reviewed 102 previously published IOS cases of all over the body, which to the best of our knowledge, is the largest review on this matter. In conclusion, although rare, IOSs should be considered as an important differential diagnosis for well-defined lytic lesions with thin sclerotic rims. On MRI, IOSs demonstrate iso-to-slightly-low signal intensity (SI) to muscle on T1WI and heterogenous high SI on T2WI with various patterns of enhancement after contrast injections. IOSs are especially important when dealing with a lesion of the mandible, sacrum, or vertebral body in middle-aged adults.

## Figures and Tables

**Figure 1 diagnostics-13-01610-f001:**
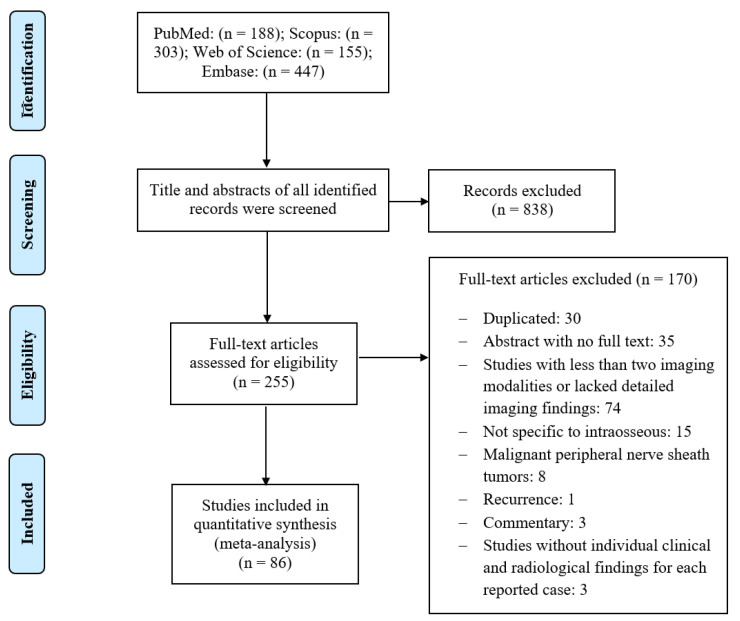
Flow chart summarizing the literature review and selection process.

**Figure 2 diagnostics-13-01610-f002:**
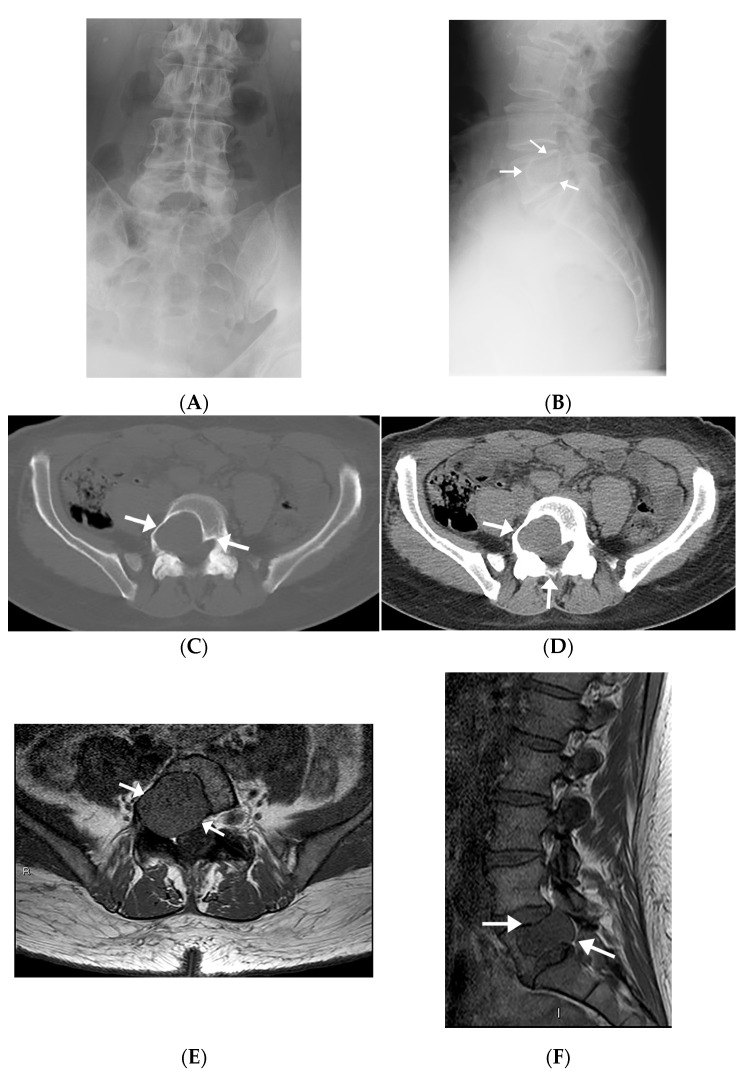
Radiographic, MR, and CT imaging appearances of intraosseous schwannoma in a 57-year-old man involving the L5 vertebral body and invading right L5 neural foramen. (**A**,**B**) The mass is occult on an AP radiograph (**A**), but the lateral radiograph (**B**) shows a large lobular lytic lesion with sclerotic margins. (**C**,**D**) Axial CT images in bone (**C**) and soft tissue (**D**) windows show an expansile lytic lesion in the right aspect of the L5 vertebral body extending into the pedicle and encroaching upon the right neural foramen and right lateral recess (arrows). (**E**,**F**) The mass extends into the right pedicle, encroaching upon the right neural foramen and right lateral recess and has a similar-to-slightly-lower signal relative to the muscle on axial (**E**) and sagittal (**F**) T1-weighted images (arrows).

**Figure 3 diagnostics-13-01610-f003:**
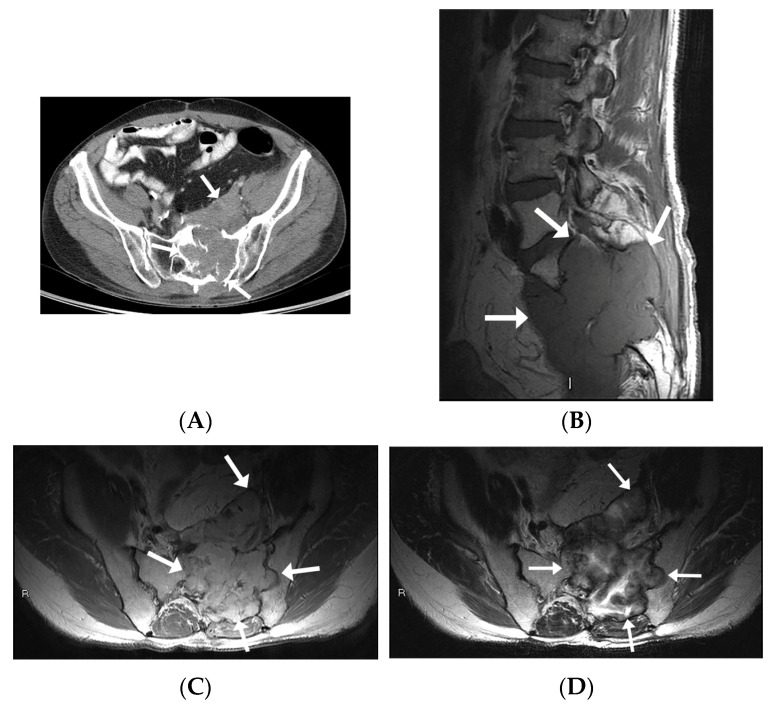
MR and CT imaging appearances of intraosseous schwannoma in a 53-year-old man involving the left sacral ala and invading S1-S3 neuroforamina, sacral epidural space, and paraspinous muscles. (**A**) Axial CT image in soft tissue window shows expansile lytic lesion with invasion into the pelvis anteriorly as well as across the posterior sacral cortex into the paraspinous muscles (arrows). (**B**) Sagittal T1-WI shows a large multi-lobulated mass involving S1, S2, and S3 vertebral bodies (arrows) with an isointense signal intensity to muscle with invasion and complete effacement of S1 through to S3 neural foramina. (**C**) Axial-contrast-enhanced T1-WI shows an enhancing mass extending anteriorly into the pelvis, posteriorly into the epidural space, as well as laterally into the left ilium (**D**) Axial T2WI shows a peripheral intermediate and central hyperintense signal.

**Figure 4 diagnostics-13-01610-f004:**
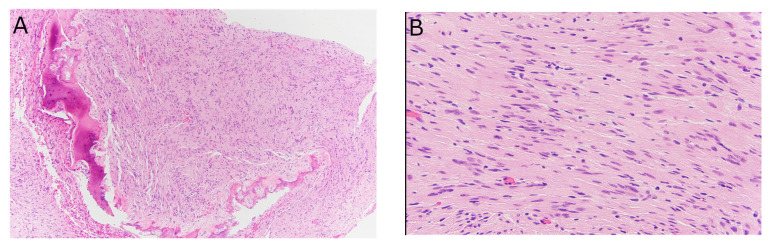
Histologic features of intraosseous schwannoma. (**A**) At the interface with the bone, there are areas of resorption and reactive change (H&E, 10×). (**B**) High-power view demonstrating Antoni A and Antoni B areas. No overt nuclear pleomorphism, mitotic activity, or other features of malignancy are identified (H&E, 40×).

**Table 1 diagnostics-13-01610-t001:** Clinical and imaging findings in patients with diagnosed IOSs.

Patient	Sex/Age	Lesion Location	Tumor Size (cm)	Clinical Findings	Radiographic Findings	CT Findings	MRI Findings
1	M/60	Left sacral ala	2.5 × 3.5 × 3.2	Right foot drop	N/A	Well-defined lytic lesion with sclerotic lobulated margins. The mass encroaches the left S1 neural foramen and abuts the nerve root with cortical destruction at the anterior margin of the vertebral body.	Avidly enhancing lesion invading S1 neural foramen and abuts exiting S1 spinal nerve root with mild mass effect. Isointense on T1WI, increased SI on STIR sequence.
2	M/51	Right sacral ala	2.1 × 1.9 × 2.8	Lower back/right lower extremity numbness, difficulty walking, and pain	Occult lesion on postoperative radiograph	Intraosseous mass extending into paraspinal soft tissues posteromedial to the right psoas muscle. Mass also extends through the cortex of right sacral wing.	Avidly enhancing lobulated lesion involving the anterior aspect of S1 vertebral body. T1WI low SI, T2WI isointense SI, and STIR high SI. Cortical breakthrough in the anterior aspect. No definite neuroforaminal involvement.
3	F/40	Sacrum midline	5.5 × 2.6 × 4.3	Severe lower back pain	Ill-defined lucent lesion, difficult to delineate from adjacent bowel gas	Lobulated lytic lesion extending into the paraspinal soft tissues posteromedial to the right psoas, separate from the exiting right L5 and S1 nerve roots.	High SI on T2WI, homogeneously low SI on T1WI, and nearly uniform enhancement. Well-defined expansile lesion involving S2 and S3 levels displacing adjacent nerve roots.
4	M/64	S1 and S2 vertebral bodies	12 × 11 × 7	Left leg pain	N/A	Heterogenous intraosseous mass with soft tissue extension, hypodense areas centrally, and calcifications in the periphery.	Large destructive mass extending into the pelvis anterior to the sacrum as well as posteriorly into the epidural space and the region of the spinous processes. Low SI on T1WI and heterogeneous SI on T2WI with areas of both high and low signal.
5	M/57	L5 vertebral body	3.8 × 3.2 × 3.5	Sudden onset right foot drop	Well-defined lytic lesion with anterior sclerotic border and cortical breakthrough posteriorly	Expansile benign appearing lytic lesion extending into the right anterior epidural space and right neural foramen with mass-effect in the thecal sac.	Heterogenous intermediate intensity on T2WI, hypointense on T1WI mass extending into the left pedicle, encroaching upon the right neural foramen and right lateral recess.
6	M/28	Left sacral ala	6.7 × 4.9 × 7	Severe low back pain	Large expansile lytic lesion with irregular sclerotic borders	Expansile lytic lesion with scalloped bony erosion involving the spinal canal from L5-S1 to S2 and extension to the left posterior pelvis.	Intensely enhancing mass with areas of central necrosis extending through the left S1 neural foramen into the pelvis. Heterogeneous SI on T2WI, isointense to muscle on T1WI.

M: male, F: female, N/A: not available, SI: signal intensity, T1WI: T1-weighted imaging, T2WI: T2-weighted imaging, STIR: short tau inversion recovery.

**Table 2 diagnostics-13-01610-t002:** Review of published cases of intra-osseous schwannoma in the head and neck.

Case	Authors, Year of Publication	Age	Sex	Clinical Findings	Lesion Location	Radiographic Findings	CT Findings	MRI Findings
1	Rozman et al., 2019, cited by [11]	68	F	Intermittent vertigo, tinnitus, facial numbness, hearing loss, diplopia, and ataxia	Petrous apex	N/A	Expansile lytic lesion extending through the petrous ridge to the apex.	Destructive lesion with displacement of the petrous and cavernous segments of the internal carotid artery. Low SI on T1-WI and high SI on T2-WI with some enhancement.
2	Sato et al., 2019, cited by [12]	35	F	Hearing disturbance and fullness	Petrous apex	N/A	Isodense lesion destructed bony structure with preserved cortex.	Isointense to brainstem on T1-WI, iso to high SI on T2-WI, with heterogeneous enhancement.
3	Tamura et al., 2015, cited by [13]	47	M	Double vision	Petrous apex	N/A	Lytic expansive lesion with enhancement, scalloped margin, and thin intact rim.	Low SI on T1-WI and slightly high SI on T2-WI with heterogenous enhancement.
4	Goiney et al., 2011, cited by [14]	48	F	New headache and decreased sensation	Petrous apex	N/A	Well-defined, lytic lesion with no extraosseous component.	Isointense to brain on T1-WI, high SI on T2-WI with enhancement.
5	Mathieu et al., 2018, cited by [15]	7	M	Painless protruding mass	Occipital bone	N/A	Well-defined, deforming intraosseous lytic lesion with sclerotic rim.	Well-defined expansile lesion. Low SI on T1-WI, high SI on T2-WI with no enhancement.
6	Goyal et al., 2008, cited by [16]	11	M	Painless swelling	Frontal bone	Lytic lesion with sclerotic rim and trabeculation.	Hypodense expansile lesion with focal cortical destruction and soft-tissue swelling. No periosteal reaction.	N/A
7	El-Bahy et al., 2004, cited by [17]	40	M	Painless proptosis and lateral gaze impairment	Sphenoorbital	N/A	Lytic lesion with sclerotic margins.	Isointense on T1-WI and hyperintense on T2-WI, and homogenous enhancement.
8	Celli et al., 1998, cited by [18]	3	M	Painless swelling	Occipital bone	Lytic lesion with sclerotic borders.	Soft tissue mass with bone erosion.	N/A
9		14	M	Painless swelling	Frontal bone	Well-defined lytic lesion with sclerotic borders.	Soft tissue mass with bone erosion.	Extradural lesion with mix SI on T1-WI, high SI on T2-WI, and peripheral intense enhancement.
10	Matsuoka et al., 2020, cited by [19]	61	M	Painless swelling	Maxilla	Well-defined lytic lesion.	Bone resorption and thinning in the left incisor region.	Well-defined, homogeneously isointense on T1-WI, and high SI on T2-WI.
11	Avinash et al., 2016, cited by [20]	38	M	Swelling	Maxilla	Partly well-defined lytic lesion that disrupted the floor of maxillary sinus.	Isodense expansile lesion with cortical perforation.	N/A
12	Oliveira et al., 2021, cited by [21]	12	F	Pain and swelling	Mandible	Lytic lesion with sclerotic border.	Hypodense expansile lesion with cortical thinning and destruction.	N/A
13	Kardouni et al., 2021, cited by [22]	24	M	Painless swelling	Mandible	Well-defined lytic lesion with a sclerotic border and root resorption.	Expansile lesion with cortical destruction and homogenous density.	N/A
14	Perkins et al., 2018, cited by [23]	22	M	Swelling	Mandible	Well-defined expansile lytic lesion with sclerotic margin and cortical erosion	Multiloculated lytic lesion with teeth root erosion and scalloping border.	N/A
15	Kargahi et al., 2012, cited by [24]	9	M	Painless swelling	Mandible	Lytic lesion with sclerotic border.	Well-defined expansile lesion with cortical thinning.	N/A
16	Suga et al., 2013, cited by [25]	33	M	Throbbing pain	Mandible	Expansive lytic lesion with sclerotic margin.	Expansive lytic fusiform lesion.	Well-defined lesion with low SI on T1-WI, high SI on T2-WI and enhancement.
17	Zhang et al., 2012, cited by [26]	35	M	Swelling and paresthesia	Mandible	Well-defined bilocular lytic lesion with sclerotic rim and dental root resorption.	Expansive lesion with bilocular destruction of the medial and lingual cortical plates.	N/A
18		39	F	Swelling	Mandible	Well-defined lytic lesion with sclerotic rim and nerve extension.	Inferior alveolar nerve involvement.	N/A
19	Agarwal et al., 2012, cited by [27]	23	F	Swelling and paresthesia	Mandible	Well-defined expansile lytic lesion with sclerotic and scalloped margin.	Unilocular, expansile, isodense lesion with marked cortical thinning and destruction.	Isointense on T1-WI, mix high SI on T2-WI and homogeneous contrast enhancement.
20	Jahanshahi et al., 2011, cited by [28]	11	F	Swelling	Mandible	Well-defined, unilocular lytic lesion with thin sclerotic borders.	Lingual cortex destruction.	N/A
21	Jiang et al., 2011, cited by [29]	39	F	Painless swelling	Mandible	Well-defined lytic lesion with sclerotic borders.	Unilocular lesion with the inferior alveolar nerve involvement.	N/A
22	Jang et al., 2009, cited by [30]	77	F	Painful swelling	Mandible	Well-defined lytic lesion with sclerotic margin.	Expansile lesion with buccal and lingual cortical thinning.	N/A
23	Gallego et al., 2009, cited by [31]	60	M	Incidental finding	Mandible	Well-defined lytic lesion with sclerotic margin.	Unilocular lesion along the inferior alveolar nerve canal.	N/A
24	Jones et al., 2008, cited by [32]	23	F	Incidental finding	Mandible	Multilocular expansile lytic lesion.	Expansible lesion to coronoid process.	N/A
25	Minowa et al., 2007, cited by [33]	67	F	Incidental finding	Mandible	Well-defined, expansive, lytic lesion with sclerotic margin.	Cortical destruction with no periosteal reaction.	Isointense SI on T1-WI and high SI on T2-WI with enhancement.
26	Kodani et al., 2003, cited by [34]	45	F	Slight diffuse swelling	Mandible	Lytic lesion with sclerotic margin.	Well-demarcated expanding mass with a clear border.	High SI on T2-WI.
27	Nakasato et al., 2000, cited by [35]	19	F	Crepitation in TM joint	Mandible	Expansive lytic lesion with sclerotic margin. No septation.	Peripheral scalloping and cortical erosion with a defect in cortex.	Isointense on T1-WI, high SI on T2-WI with peripheral enhancement.
28	Park et al., 1999, cited by [36]	29	F	Incidental finding	Mandible	Well-defined lytic lesion with sclerotic margin.	Cortical thinning but no destruction.	N/A
29	Belli et al., 1997, cited by [37]	8	M	Painless swelling	Mandible	Well-defined lytic lesion with thin septa.	Enhanced lesion with cortical reabsorption but no destruction or periosteal reaction.	N/A
30	Xu et al., 2020, cited by [38]	56	M	Chronic and persistent neck pain	C7 vertebra	Lytic lesion with sclerotic border and C7 vertebral body height loss.	Lytic bony lesion with well-preserved vertebral cortex.	Well-defined low SI on T1-WI, high SI on T2-WI. Anterior cortex destruction. No spinal cord extension.
31	Mohanty et al., 2012, cited by [39]	10	M	Painless swelling	C4 vertebra	Lytic with sclerotic border and anterior cortical destruction	Lytic lesion with soft tissue component and no intraspinal extension.	Low SI on T1WI, mix SI on T2WI, and uniform enhancement.
32	Peng et al., 2011 cited by [40]	44	M	Progressive dizziness and mild weakness in the right upper limb	C3 vertebra	C3 vertebral collapse, loss of the right C3 pedicle and C3 to C4 disc space, enlargement of intervertebral foramen, and displacement of C2 and C3 vertebrae anteriorly	Expansile, lytic, and invasive lesion destroying the right pedicle, laminar, and spinal process. A soft tissue mass extending from the C3 vertebral body to the right spinal canal and paravertebral.	T2-WI showed an inhomogeneously hyperintense mass involving in the C3 and partial C2 vertebral body, displacing the spinal cord, and extending the spinal cord posteriorly
33	Mizutani et al., 2010, cited by [41]	44	F	Incidental finding	C4 vertebra	N/A	Bony defect with preserved cortex.	Well-defined isointense lesion on T1-WI, high SI on T2-WI, and homogeneous enhancement.
34	Nannapaneni et al., 2005, cited by [42]	42	M	Incidental finding	C5 vertebra	Lytic lesion with C4-5-disc space and C5 vertebral body height loss with soft tissue shadow.	Expansile lytic lesion with sclerotic margin, faintly calcified, and poorly enhanced with cortical destruction.	Well-defined isointense mass on T1-WI, high SI on T2-WI with marginal enhancement and exophytic component displacing carotid sheath.
35	Schreuder et al., 2001, cited by [43]	38	F	Neck pain and dysphagia	C6 vertebra	Lytic with sclerotic borders and anterior cortical destruction.	N/A	Soft tissue displacement with no invasion. High SI on T1 and T2-WI with fat and high fluid content.

M: male, F: female, N/A: not available, SI: signal intensity, T1WI: T1-weighted imaging, T2WI: T2-weighted imaging, TM: temporomandibular.

**Table 3 diagnostics-13-01610-t003:** Review of published cases of intra-osseous schwannoma in trunk.

Case	Authors, Year of Publication	Age	Sex	Clinical Findings	Lesion Location	Radiographic Findings	CT Findings	MRI Findings
1	Mizuno et al., 2010, cited by [44]	38	F	Incidental finding	Sternum	N/A	Well-defined low-density nodule expanded into the thoracic cavity.	Isointense on T1-WI and a high SI on T2-WI with no enhancement.
2	Takata et al., 1999, cited by [45]	10	F	N/A	Sternum	Lytic lesion with sclerotic margin	Mass with cortical destruction and soft tissue extension.	Isointense SI on T1-WI and high SI on T2-WI with no invasion to thoracic cavity.
3	Lee et al., 2019, cited by [46]	58	M	Painful mass	Seventh rib	Expansile lytic lesion	A mass with peripherally curved calcification and cortical destruction resulting in soft tissue mass formation.	N/A
4	Nguyen et al., 2017, cited by [47]	48	M	Incidental finding	Scapula	Expansile lytic lesion with sclerotic borders.	Lytic lesion with thin-rimmed cortical bone remodeling and lack of internal calcifications.	Contrast-enhancing, lobulated lesion on T1-WI with no adjacent soft tissue involvement.
5	Tian et al., 2014, cited by [48]	42	F	Shoulder pain	Scapula	Lytic lesion with sclerotic margin	Destructive mass extending into surrounding soft tissues.	Low to intermediate SI on T1-WI, high SI on T2-WI and destruction of left scapula and glenoid.
6	Reyniers et al., 2021, cited by [4]	49	F	Shoulder pain and paresthesia	Glenoid	Well-defined lytic lesion with thin sclerotic rim and trabeculation.	N/A	Expansile lesion with cortical thinning and focal breach. Isointense on T1-WI, high SI on T2-WI, and heterogenous enhancement.
7	Zaidman et al., 2019, cited by [49]	56	F	Incidental finding	T1 vertebra	N/A	Lytic lesion with no sclerosis.	High enhancement and high SI on T2-WI, without extension to surrounding soft tissue.
8	Kojima et al., 2011, cited by [50]	60	M	Pain, gait disturbance, numbness, weakness	T9 vertebra	N/A	Large lytic lesion with erosion of the lamina, spinous process, and cortical destruction.	Isointense on T1-WI, high mixed SI on T2-WI and irregular enhancement with paravertebral muscles extension.
9	Zhang et al., 2015, cited by [51]	54	F	Gait disturbance, paresthesia,	T9 vertebra	N/A	Bone erosion and destruction.	Isointense on T1-WI, mixed SI on T2-WI. Extending to spinal canal and paravertebral areas.
10		71	M	Pain, gait disturbance, paresthesia,	L4 vertebra	N/A	Lytic lesion with sclerotic rim and severe vertebral destruction.	Isointense on T1-WI, heterogeneous on T2-WI, irregular enhancement.
11	Choudry et al., 2007, cited by [52]	18	M	Back pain, and weakness	T12 vertebra	Gross cystic changes with partial collapse of vertebral body.	N/A	Perivertebral protrusion compressing the thecal sac and neural foramina. Low SI on T1-WI, heterogenous SI on T2-WI and no enhancement.
12	Nooraie et al., 1997, cited by [53]	46	M	Burst fracture following car accident	T12 vertebra	Lytic lesion with a sclerotic rim in the T12 vertebra.	Large lytic lesion, involving all three spinal columns, with sclerotic rim, pedicle erosion, and cortical destruction.	N/A
13	Park et al., 2009, cited by [54]	48	F	Back pain and weakness	L4 vertebra	N/A	Expansile lytic tumor with sclerotic rim.	Peripheral enhancement of the degenerative portion with vertebral body pathologic fracture. High SI on T2-WI and isointense on T1-WI.
14	Chang et al., 1998, cited by [55]	58	M	Numbness, pain, and weakness	L4 vertebra	Expansile lytic lesion with smooth and sclerotic border.	Hypervascular mass with cephalad and caudal extension causing posterior compression of the thecal sac.	L4 and L5 vertebral body mass with thecal sac and bilateral neuroforamina compression.
15	Song et al., 2014, cited by [56]	44	M	Low back pain with radiation	L5 vertebra	N/A	Irregular lytic lesion with marginal sclerosis caused isthmic spondylolysis. Posterior protrusion with cortical destruction, and thecal sac compression.	Low SI on T1-WI and high SI on T2-WI. Mild heterogeneous enhancement.
16	Youn et al., 2012, cited by [57]	65	M	Progressive lower back pain	L2 vertebra	Expansile lytic lesion with sclerotic margin.	Lytic lesion with a sclerotic margin.	High mixed SI on T2-WI corresponding to cystic degeneration, isointense on T1-WI, and irregular enhancement.
17–29	Summers et al., 2018, cited by [5]	25–80	7 M 6 F	Low back pain (6/13), Radiculopathy (2/13), incidental (4/13), not specified (7/13)	Sacrum	N/A	Solid, expansile lytic lesions with sclerotic margins, pathologic fracture in 1/13.	T1-WI: Hypointense (5/8) Isointense (3/8), T2-WI: Heterogenous (8/8) solid (4/8) or cystic lesions with fluid-fluid levels (4/8) that may exhibit target sign. Post-contrast images: Heterogeneous enhancing solid component (8/8), non-enhancing cysts (4/8), and no necrosis.
30	Aaron et al., 1995, cited by [58]	53	M	Back pain and paresthesia	Midline of sacrum	Lytic with thin sclerotic border.	Lucent homogenous lesion without calcification abutting the paraspinal musculature.	Solitary isointense mass on T1-WI, moderately high SI on T2-WI and marked enhancement.
31	Takeyama et al., 2001, cited by [59]	45	M	Pain and claudication	Sacrum	Lytic lesion with cortical destruction.	Gigantic retroperitoneal mass displacing the bladder ventrally.	Sacral lesion with cranial expansion of tumor. High SI on T2-WI.
32	Silva et al., 2019, cited by [60]	22	M	Lumbar pain and difficulty walking	Sacrum	N/A	Lytic lesion with soft tissue component and sacral wing scalloping.	Contrast-enhancing lesion.
33	Mutlu et al., 2015, cited by [61]	38	M	Progressive pain and claudication	Sacrum	N/A	A soft tissue mass with scalloping of the surrounding bone.	Obliteration of sacral canal. Isointense on T1-WI, and hyperintense on T2-WI.
34	Kato et al., 2015, cited by [62]	27	M	Incidental finding	Ilium	N/A	Well-demarcated lytic lesion with punctate calcification, sclerotic rim and cortical erosion. Heterogenous enhancement.	Heterogeneously high SI on T2-WI with peripheral hypointense rim. Low SI on T1-WI and heterogenous enhancement.
35	Benazzo et al., 2013, cited by [63]	63	F	Paroxysmal pain	Iliopubic ramus	Well-defined expansile lytic lesion with cortical destruction	N/A	Expansile lesion with extraosseous extension. Low SI on T1-WI, and high SI on T2-WI with heterogenous enhancement.

M: male, F: female, N/A: not available, SI: signal intensity, T1WI: T1-weighted imaging, T2WI: T2-weighted imaging, STIR: short tau inversion recovery.

**Table 4 diagnostics-13-01610-t004:** Review of published cases of intra-osseous schwannoma in extremities.

Case	Authors, Year of Publication	Age	Sex	Clinical Findings	Lesion Location	Radiographic Findings	CT Findings	MRI Findings
1	Kamath et al., 2021, cited by [64]	45	F	Fracture following low impact fall	Humerus	Lytic lesion with pathological fracture	N/A	Low SI on T1-WI and high SI on T2-WI/STIR with few septations and cortical thinning. No cortical erosions.
2	Huajun et al., 2021, cited by [65]	55	F	Fracture following low impact fall	Humerus	Well-defined lytic with sclerotic margin, endosteal scalloping. No calcifications	Periosteal elevation and pathologic fracture.	Cortical invasion with associated soft tissue edema. Isointense on T1-WI, and high SI on T2-WIs.
3	Mutema and Sorger, 2002, cited by [1]	33	M	Pain, swelling and reduced ROM	Humerus	Expansile, lytic lesion with sclerotic margin, endosteal scalloping and no cortical destruction	N/A	Low SI on T1-WI and high SI on T2-WI associated with soft tissue edema.
4	Baǧci et al., 2010, cited by [66]	19	F	Pathologic fracture	Radius	Expansile lytic lesion with sclerotic rim	N/A	Intramedullary mass with soft tissue component and no homogeneous pattern.
5	Giné et al., 1999, cited by [67]	45	M	Painless swelling	Radius	Lytic lesion with sclerotic borders, trabeculation and cortical destruction	Destruction of the dorsal cortex with associated soft tissue component.	Hyperintense mass on T1-WI and hypointense on T2-WI.
6	Lim et al., 2021, cited by [68]	77	F	Fracture following low impact fall	Ulna	Well-defined lytic lesion with a thin sclerotic rim, pathological fracture	N/A	Expansile intraosseous mass invading the adjacent muscle and subcutaneous fat tissue.
7	Suzuki et al., 2016, cited by [69]	87	F	Rapidly-growing mass with tenderness	Ulna	Well-defined, lytic lesion with sclerotic margin, pathological fracture, and cortical thinning	Cortical expansion and thinning with no calcification.	Isointense on T1-WI, and heterogenous high SI on T2-WI with soft tissue extension.
8	Kito et al., 2014, cited by [70]	21	F	Pain	Ulna	Well-defined, lytic expansile with marginal sclerosis and trabeculation	Cortical destruction with associated soft tissue mass. No periosteal reaction.	Isointense mass on T1-WI, heterogeneously hyperintense on T2-WI, and uniform enhancement.
9	Gurkan et al., 2017, cited by [71]	34	F	Pain and mild swelling	Hamate	Well-defined lytic lesion with sclerotic borders and cortical destruction. No calcifications	Expansile lytic lesion with dorsal cortical breakthrough. No periosteal reaction.	Cortical disruption with soft tissue extension. Hyperintense on PD-fat sat and T2-WI sequences and homogenous enhancement.
10	Vora et al., 2000, cited by [72]	45	M	Pain with no swelling	Metacarpal	Expansile lytic with pathologic fracture and periosteal reaction. No sclerosis or calcification	N/A	Homogeneously low SI on T1-WI and high SI on T2-WI with endosteal scalloping and periosteal reaction.
11	Afshar and Afaghi, 2010, cited by [73]	12	M	Painless swelling	Metacarpal	Well-defined, lytic expansile lesion with sclerotic margin and cortical disruption	N/A	Cortical disruption and marrow replacement with heterogeneous SI on T1-WI and high SI on T2-WI.
12	Verma et al., 2002, cited by [74]	38	M	Pain and insidious weakness	Femur	Smooth scalloping of the outer aspect of the medial cortex with no matrix mineralization	N/A	Well-defined isointense mass on T1-WI, hyperintense on both T2-WI and STIR, with enhancement and no cortical breaching.
13	Hoshi et al., 2012, cited by [75]	44	F	Pain	Femur	Well-defined lytic with sclerotic margin at the lesser trochanter.	Osteolytic with cortical ballooning and thinning, and periosteal reaction.	Expansile low- to iso-intensity on T1-WI and heterogeneously high SI on T2-WI with marked enhancement.
14	Wang et al., 2014, cited by [10]	42	M	Pain	Femur	Well-defined expansile lytic with sclerotic margins and cortical destruction	N/A	Heterogeneously isointense on T1-WI, and hyperintense on FSE-T2-WI, with homogenous enhancement.
15	Al-Lhedan, 2017, cited by [76]	18	F	Painless swelling	Femur	Expansile lytic lesion with sclerotic margins	N/A	Low SI on T1-WI and high SI on fluid sensitive sequences with avid enhancement and soft tissue component.
16	McAleese et al., 2020, cited by [77]	55	F	Pain	Femur	N/A	Lytic lesion with a thin sclerotic margin causing mild cortical scalloping.	Well marginated expansile lesion with a homogeneous consistency. Low SI on T1-WI, high SI on T2-WI.
17	Mardi et al., 2021, cited by [78]	46	F	Pain and swelling	Tibia	Expansile lytic lesion with sclerotic margin and trabeculation	N/A	Well-defined lobulated eroding mass with thin septa.
18	Kashima et al., 2013, cited by [79]	55	F	Leg pain	Tibia	Well-defined lytic lesion with a thin sclerotic rim	N/A	Isointense on T1-WI and high SI on T2-W fat-suppressed sequence.
19		62	F	Fracture following low impact fall	Tibia	Pathological fracture through well-defined lytic lesion with thin sclerotic rim	N/A	Dumbbell-shaped intramedullary lesion with the waist of the dumbbell centered on the cortex.
20	Wang et al., 2014, cited by [80]	75	F	Painful small lump	Tibia	Well-defined lytic lesion with a sclerotic rim. No calcification or periosteal reaction	Lobulated intraosseous lesion with cortical breakthrough.	N/A
21	Meyer et al., 2008, cited by [81]	13	M	Increasing pain	Tibia	Well-defined lytic lesion with sclerotic rim and intact growth plate	Lytic lesion cortical thinning and destruction. No calcification or periosteal reaction	Intra-articulaire expansion. Isointense on T1-WI, heterogeneous high SI on T2-WI and intensive enhancement.
22	Ilgenfritz et al., 2006, cited by [82]	34	F	Pain and asymmetry	Tibia and fibula	Lytic lesions with well-defined sclerotic margins and periosteal reaction	Cortical destruction and with eroding soft tissue mass.	Isointense on T1-WI and heterogenous hyperintense on T2-WI.
23	Palocaren et al., 2008, cited by [83]	14	M	Pain and swelling of the right leg	Fibula	Well-defined lytic expansile lesion with sclerotic rim, trabeculations, and cortical destruction	Cortical expansion and destruction with soft tissue extension.	Isointense on T1-WI and homogeneous high SI on T2-WI with surrounding soft-tissue edema. No fluid levels.
24	Aoki et al., 1997, cited by [84]	56	F	Right knee pain	Fibula	Well-defined lytic lesion with marginal sclerosis and calcifications	Cortical expansion and destruction with partial discontinuity, no soft tissue extension.	Isointense SI on T1-WI and higher SI than subcutaneous fat on T2-WI. Uneven enhancement with avidly enhancing nodules.
25	Drumond et al., 2020, cited by [85]	49	M	Pain and swelling	Calcaneus	Well-delimited lytic lesion with sclerotic rim and cortical rupture. No calcification	N/A	Low SI on T1-WI, heterogeneous SI on T2-WI, and intense enhancement. Cortical rupture with extensive soft tissue component.
26	Haberal et al., 2018, cited by [2]	35	F	Right heel pain	Calcaneus	Cystic lytic with sclerotic rims	N/A	Cystic hypointense lesion on T1-WI and hyperintense on T2-WI with cortical involvement and well-defined borders.
27	Sochart et al., 1995, cited by [86]	23	F	Intermittent pain and swelling	Calcaneum	Cystic lesion with sclerotic margins and no calcification	Attenuation and thinning of the lateral cortex.	N/A
28	Pyati et al., 1996, cited by [87]	35	F	Painful heel caused limping	Calcaneus	Multiloculated, expansile lytic lesion	Large lytic lesion with medial wall erosion.	N/A
29	Flores Santos et al., 2014, cited by [88]	70	F	Visible mass with pain and swelling	Cuboid	Lytic lesion with sclerotic margins and cortical disruption	Large soft tissue mass with extensive bone involvement.	Isointense mass on T1-WI along the posterior tibial nerve. Hyperintense signal on T2-WI.
30	Wang et al., 2016, cited by [89]	50	F	Swelling with dull pain	Foot	Well-defined lytic lesion with sclerotic margin and trabeculation	Bone destruction with extension to subcutaneous tissue. No calcifications.	Isointense to skeletal muscle on T1-WI and hyperintense to subcutaneous fat on T2-WI.
31	Ansari et al., 2014, cited by [90]	48	F	Swelling	First metatarsal	Well-defined expansible lytic lesion with sclerotic margins	N/A	Isointense on T1-WI, high SI on T2-WI, with cortical break.
32	Meek et al., 2007, cited by [91]	27	F	Pathologic fracture	Metatarsal	Lytic lesion with sclerotic margins	N/A	Amorphous mass in medullary canal with cortical destruction.

M: male, F: female, N/A: not available, SI: signal intensity, T1WI: T1-weighted imaging, T2WI: T2-weighted imaging, ROM: range of motion, STIR: short tau inversion recovery, FSE: Fast spin echo.

## Data Availability

This article does not contain any new generated or analyzed data.

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
