# Peer review of "Imaging Features of Intraosseous Schwannoma: A Case Series and Review of the Literature"

_diagnostics, 2023, doi:10.3390/diagnostics13091610_

Round 1

Reviewer 1 Report

I commend the authors for their research entitled »Imaging and Clinical Features of Intraosseous Schwannoma; A Case Series and Review of Literature«. The manuscript focused on clinical and imaging features of patients with pathologically confirmed intraosseous schwannoma. A thorough additional literature search resulted in 102 published cases with the same imaging characteristics. The topic is very interesting, the manuscript is clearly written and the discussion is relevant to the results. However, the conclusions are scarce and represent the worst part of the manuscript. Therefore, it should be rewritten. Please state clearly what is making your research so unique. To the best of my knowledge, no article published so far has covered these rare lesions so meticulously.  

Author Response

Authors response:   

Thank you for your comment. We agree with your comment. We revised the conclusion to improve it and make it concise. We hope the new version meets your expectations. 

Authors action:   

We have changed the conclusion in page 23, line 370 accordingly: “We evaluated 6 cases of sacral intraosseous schwannoma and reviewed 102 previously published IOS cases of all over the body, which to the best of our knowledge is the largest review on this matter. In conclusion although rare, IOSs should be considered as an important differential diagnosis for well-defined lytic lesions with thin sclerotic rims. On MRI, IOSs demonstrate iso to slightly low signal intensity (SI) to muscle on T1WI and heterogenous high SI on T2WI with various patterns of enhancement after contrast injection. IOSs are especially important when dealing with a lesion of the mandible, sacrum, or vertebral body in middle-aged adults.” 

Reviewer 2 Report

The article describes a case series of intraosseous schwannomas and provides a literature review of the previously described cases.

This is a rare pathology that is often not included in the differential diagnosis of osteolytic lesions. I think that a comprehensive review focusing on this clinical entity would be useful to the literature. The idea is a valid one in my opinion. The article is well written in all its parts: the introduction concise and informative, the methods adequately described, the discussion detailed, the conclusions consistent with the results. 

I only have a few considerations for the authors. The literature review and discussion are strongly oriented towards radiological aspects, so I would change the title to 'Imaging Features of Intraosseous Schwannoma'. Nevertheless, in the description of the original case series, I think it would be useful to include a subsection focused on the clinical aspects: treatment, complications, recurrences, eventual treatment of recurrences.

Thank you.

Author Response

Authors response:   

Thank you for your comment. We agree with your comment. Although unfortunately we don’t have access to the clinical information of our own cases and most of the previously reported cases also did not provide this information in their article. Therefore, we changed the title to explain that the focus of this paper is on the imaging finding of IOSs. 

Authors action:   

We have changed the title accordingly: “Imaging Features of Intraosseous Schwannoma; A Case Series and Review of Literature” 

Reviewer 3 Report

It's a goof paper on a rare diagnosis but it's interesting to review this cases

Author Response

Authors response:   

Thank you for your comment. We appreciate your time to review our manuscript.